# The Anatomy of the Circle of Willis Is Not a Strong Enough Predictive Factor for the Prognosis of Cross-Clamping Intolerance during Carotid Endarterectomy

**DOI:** 10.3390/jcm9123913

**Published:** 2020-12-02

**Authors:** Piotr Myrcha, Andrzej Lewczuk, Maciej Jakuciński, Mariusz Kozak, Dawid Siemieniuk, Dorota Różański, Dariusz Koziorowski, Witold Woźniak

**Affiliations:** 1Chair and Clinic of General and Vascular Surgery, Faculty of Medicine, Medical University of Warsaw, W02-091 Warsaw, Poland; drww@onet.pl; 2Department of General, Vascular and Oncological Surgery, Masovian Brodnowski Hospital, 03-242 Warsaw, Poland; kozakmariusz88@gmail.com (M.K.); dawid4310@gmail.com (D.S.); 3Department of General Surgery, Medicover Hospital, 00-807 Warsaw, Poland; lewczuk78@poczta.fm; 4Department of Radiology, Mazowiecki Brodnowski Hospital, 03-242 Warsaw, Poland; mjakucinski@gmail.com; 5Department of Neurology, Faculty of Health Science, Medical University of Warsaw, 02-091 Warsaw, Poland; dvd@mail.com (D.R.); dkoziorowski@wum.edu.pl (D.K.)

**Keywords:** carotid endarterectomy, carotid magnetic resonance imaging, magnetic resonance perfusion, cerebrovascular disease/stroke, cross-clamping intolerance

## Abstract

Carotid endarterectomy (CEA) is safe and effective in reducing the risk of stroke in symptomatic severe carotid artery stenosis. Having information about cross-clamping (CC) intolerance before surgery may reduce the complication rate. The purpose of this study was to assess the usefulness of magnetic resonance angiography (MRA) and magnetic resonance angiography perfusion (P-MR) in determining the risk of CC intolerance during CEA. Material and methods: 40 patients after CEA with CC intolerance were included in Group I, and 15 with CC tolerance in Group II. All patients underwent MRA of the circle of Willis (CoW), P-MR with or without Acetazolamide; P(A)-MR in the postoperative period. Results: CoW was normal in the MRA in three cases (7.5%) in Group I, and in eight (53%) in Group II. We found P-MR abnormalities in all patients from Group I and in 40% from Group II. Using a calculated cut-off point of 0.322, the patients were classified as CC tolerant with 100% sensitivity or as CC intolerant with 95% specificity. After evaluating P-MR or MRA alone, the percentage of false negative results significantly increased. Conclusion: The highest value in predicting cross-clamping intolerance is achieved by using analysis of P(A)-MR and MRA of the CoW in combination.

## 1. Introduction

Neurological complications during and after internal carotid artery (ICA) endarterectomy (CEA) occur as a result of intraoperative embolization, hypoperfusion during the ICA cross-clamping (CC), intracranial hemorrhage or hyperperfusion syndrome [1,2].

The risk of intraoperative neurological complications is particularly high in patients who do not tolerate CC, and extremely high in the case of acute thrombosis of the operated artery or repeated operations. Predicting the occurrence of CC intolerance should affect the operational technique and the selection of a surgical team experienced in using shunts [3].

The methods used to evaluate the anatomy of the circle of Willis led to too many false-positive results [4]. As a consequence, patients are unnecessarily selected for a shunt or for carotid artery stenting (CAS) even if they have contraindications.

The purpose of this study was to preoperatively assess the risk of cerebral ischemia during carotid CC using different diagnostic tests. Our goal was also to investigate factors affecting cerebral ischemia during carotid CC.

## 2. Experimental Section

Patients who presented ischemic symptoms within 2 to 60 s after CC, and the operation was cancelled, were included in the study (Group I). A group of patients without symptoms of cerebral ischemia after CC that underwent CEA, was studied as controls (Group II).

Preoperatively, carotid ultrasound examination (DUS), magnetic resonance angiography (MRA) of the extracranial and intracranial arteries and magnetic resonance brain perfusion tests (P-MR) with and without Acetazolamide (PA-MR) were performed on all patients.

The DUS study was performed using a Siemens (Munchen, Germany) ultrasound machine with the use of both linear (5.1–9 MHz) and convex (2.8–5.1 MHz) probes. The level of ICA stenosis was assessed according to the NASCET criteria.

A GE Healthcare (Chicago IL, USA) Signa Excite 1.5 T, 8-channel magnetic resonance scanner was used to perform MRA and P-MRA. Brain perfusion was assessed using Functool (GE Healthcare) 2-MR Standard software on a GE Advantage Workstation.

In order to evaluate the intracranial arteries, MRA was performed using the time of flight (TOF) technique with 3D reconstruction. The posterior communicating arteries (PCoA) were examined by performing T2-dependent images in the FSE sequence (slice thickness 1 mm).

P-MR was performed after administration of a contrast, Prohance 2–3 mmol/kg, by an automatic syringe at a rate of 5 mL/s. Venflon (1.2–1.4 mm) was used for the administration of the contrast which was put into the ulnar vein in the area of elbow flexion. The subject of the analysis was the difference in cerebral perfusion parameters expressed in the mean time of flow, in the cases of the basic examination and the PA-MR. Differences of up to 10% in brain perfusion between similar region of interest (ROI) points of the symmetrical structures of the hemispheres were considered a normal variant.

The arterial input function (AIF) was determined for the area of the middle cerebral artery (MCA). Values from perfusion measurement points in symmetric structures of the brain (region of interest—ROI) were used to calculate the parameters of the cerebral circulation—mean transit time (MTT).

In the brain perfusion images, the difference in the flow between the analogous ROI points of the symmetrical structures of the left and right hemispheres was determined: semiovale centre—near the middle furrow; the white matter of the frontal lobes—anterior horns of the ventricular system; white matter of the occipital lobes—posterior horns of the ventricular system; hill; the hemisphere of the cerebellum. Perfusion maps were compared in the baseline study (first day) and in the PA-MR (second day). Two independent diagnostics evaluated the results. SPSS package ver.14.0 (IBM, Armonk, NY, USA) was used for the statistical calculations.

In order to search for variables related to tolerance to CC, student’s *t* tests were performed. For the purposes of the study, the W index was created, derived from the logistic model, which determines the measurement of intolerance to CC depending on the number and type of pathology in the circle of Willis.

During the analysis of MRA results, a numerical value from 0 to 2 was assigned to each artery (0—no pathology, 1—flow limitation, 2—no flow). In addition, a developmental variant in the form of carotization (PCoA as a branch of ICA) of the posterior cerebral arteries (PCA) was considered. Depending on the occurrence of the above variant, 1 or 2 points were added. The sum of all values was used to express the measurement of intolerance to CC (the higher the W index, the higher the probability of intolerance to CC).

An optimal model predicting the probability of obtaining a positive result of the dependent variable resistance to CC is described by the following formula:(1)$$P\;\left(\mathrm{CC}\;\mathrm{tolerance}\right)\;=\;\frac{1}{1+e^{34.98+1.063\times \mathrm{DA}+1.906\times W\;\mathrm{Index}-26.53\times \mathrm{HT}}}$$

DA—difference Acetazolamide. Difference in the flow between the corresponding ROI points of the symmetrical structures of the left and right hemispheres of the brain, expressed as a percentage (we need only substitute a numerical value in the formula), W index—considered the arteries 1–7 listed in the diagram, HT—hypertension. “1” In the absence of hypertension and “2” in the presence of hypertension (Figure 1).

We obtained a numerical value in the range from 0 to 1. The higher the value, the higher the probability of CC tolerance. The receiver operating characteristic (ROC) curve is a tool for assessing the accuracy of the classifier and supports the decision system under conditions of uncertainty.

The ROC analysis was performed to create an algorithm in a controlled manner, based on which it will be possible to determine the efficiency of CC (Figure 2).

Using a cut-off point equal to or greater than 0.322, the patient can be correctly assigned to one of two groups. TP (Group II) with CC tolerance, sensitivity = 100%, and TN (Group I) with CC intolerance, specificity = 95%. The accuracy of classification (ACC) was 96.2%.

## 3. Results

A total of 40 patients, 14 women and 26 men between the ages of 51 and 84 (average of 68.1), were included in this study (Group I); 27.5% of the patients were asymptomatic. All had ipsilateral ICA stenosis > 70% and underwent an attempt at CEA. All 40 patients developed ischemic symptoms during CC with withdrawal from the elective CEA (Group I). The time of appearance of neurological symptoms during CC ranged from 2 to 60 s (average 21 s). These patients represented 4.8% of the 833 CEAs performed during the same period at our institution.

The control group (Group II) included 15 patients with cross-clamping tolerance, operated on by a single surgeon (P.M.), who gave their informed consent to perform pre- and postoperative examinations (lack of funding limited the number of patients entered in the study). In this group were 4 women and 11 men, aged 66 to 83 (average 69.8); 26.6% of the patients were asymptomatic. At least one comorbidity occurred in 54 patients (98.2%) (Table 1).

In both groups, DUS study on the operated side showed ICA stenosis above 70%. There weren’t any occlusions or near-occlusions.

The contralateral ICA was occluded in six patients from Group I. Stenosis in the range of 0–49% was in, 50–69% in 22 and above 70% in 6 patients. In Group II; 0–49% stenosis was found in two, 50–69% in nine, above 70% in two and occlusion in two patients.

### 3.1. Results of MRA

The circle of Willis had a normal anatomy in three patients from Group I (7.5%) and in eight from Group II (53%). Other patients from both groups had single or multiple abnormalities (Figure 3).

The only pathology found in Group II was bilateral (three patients) or unilateral (five patients) lack of posterior communicating artery (PCoA). In Group I the first segment of the anterior cerebral artery (A1) was absent in 21 patients (52.5%). Twelve patients (30%) had decreased A1 signal. In 12 cases (30%) the anterior communicating artery (ACoA) was absent (Table 2). In 29 cases (72.5%) no flow in the PCoA was observed. One patient had no signal in the first segment of the posterior cerebral artery (P1). Among five patients from Group I, carotisation was observed.

### 3.2. Results of the P-MR and PA-MR

In Group I, 24 patients (60%) had abnormalities in cerebral perfusion parameters on P-MR without Acetazolamide. In 15 patients there was a reduction in perfusion in the hemisphere on the affected side. In PA-MR, perfusion disorders were found in all subjects. In 28 cases, they appeared on the affected side (Figure 4).

In Group II in one patient (6.6%), after the basic P-MR, perfusion disorders on the operated side were observed. In six cases (40%) with PA-MR the differences between the hemispheres were greater than 10%. They corresponded to the operated side.

Statistically significant results indicating the relationship between the variable tolerance to CC and potential predictors of tolerance to CC are presented in Table 3.

There were 14 student’s t tests counted in the whole sample. The independent variable is the resistance to CC, measured on two levels: positive CC (intolerance) and negative CC (tolerance). All tests were carried out at a significance level of 5.0%.

A cut-off point for the true positive rate group of patients with a prediction value (P) equal to or greater than 0.322 was marked. The cut-off value was based on a set false positive rate of 5.0%—the maximum prediction error of CC tolerance (Table 4).

Depending on the range of data analyzed, (model using PA-MR + MRA data, model using only MRA data, model using only PA-MR data), the predictive power of the indicators of tolerance to CC differ significantly. The difference in the power of prediction can be expressed by several variables. The easiest and the most abstract one is R^2^ factor. However, a much better criterion is the assessment of the ROC curve. The ACC criterion is also a practical criterion (Table 5).

The best predictive power indicators are obtained from the analysis of PA-MR and the MRA analysis in combination. This is also confirmed by data from ROC analyses: the error of incorrectly predicting CC intolerance (false negative—FN) was 0% while the information obtained from both the PA-MR and MRA were used. However, based only on the PA-MR the FN error increased to 15.4%, and based only on the MRA study the risk of error was 45.4%.

## 4. Discussion

CC intolerance is influenced by many factors, including demographic and epidemiological [5]. Age, sex, hypertension, diabetes and occlusion of the remaining extracranial arteries may all increase the risk of cerebral ischemia during clamping. However, their occurrence does not determine lack of tolerance to CC [6]. A factor such as abnormalities of the circle of Willis may be congenital. It has an unpredictable effect on CC, therefore we decided to extend the diagnosis with PA-MR [7,8].

The combined analysis of the results of PA-MR and MRA showed that the most important predictor of tolerance to CC is the combined result of PA-MR and MRA of intracranial arteries, and also the W index calculated on their basis. The third variable significantly related to CC tolerance is hypertension.

The analyses conducted did not show a significant effect on resistance to cross-clamping of factors such as contralateral stenosis of ICA and the presence of neurological symptoms in the preoperative period. The study groups were small. We treat the results as a trend and understand that a large cohort would be necessary.

The analysis of the data enabled the derivation of the formula shown earlier by which the measurement of the effects of CC (probability) can be determined.

We wanted to determine the two classes of the patients (CC tolerance vs. CC intolerance) before the planned operation. The result of the ROC analysis was used to determine the cut-off point. The ROC model assumes that the error of incorrect prediction of condition (FP) will not exceed 5%. After taking this assumption into consideration in regard to the results of PA-MR and MRA, the cut-off point was determined to have the value of 0.322. In practice, the use of this value means that if the value of *p* (obtained from the formula given above) is equal to or greater than 0.322, then the risk of positive clamping is minimal.

The methods used to predict cerebral ischemia during CEA can be divided into two groups. The first one assesses this risk intraoperatively using methods ranging from the simplest, such as the assessment of the neurological status of patients operated under regional anesthesia or the assessment of the backflow pressure [9], to those requiring the use of complicated apparatus and experience in the interpretation of the results obtained. In day-to-day practice, transcranial Doppler [2], electroencephalography (EEG) [10,11], somatosensory evoked potentials (SSEP) [12], near-infrared spectroscopy (NIRS) [13] or cerebral oximetry are used [14]. Provocative occlusion testing (the Matas test) is rarely used due to the risk of embolic complications, which is true in the MR version as well [15,16]. Evaluation of the results of these tests influences the decision to apply a shunt or not. AbuRahma et al. [17], on the basis of prospective studies, found that there are no significant differences in CEA results in patients in whom a shunt is used routinely or selectively. The literature review conducted by the same author produced similar results—there is only a little difference in outcome of routine shunting and selective shunting (1.4% vs. 1.6% stroke rate) [18]. This may mean that there is a subset of patients who are at risk of intraoperative stroke even with shunt use, e.g., those with extreme collateral insufficiency.

Such a complication rate is of course acceptable in symptomatic patients. Advances in best medical treatment may mean that in asymptomatic patients this percentage of neurological complications will be too high. Excluding patients with potential cross-clamping intolerance from surgical treatment and offering them, for example, stenting may further reduce this complication rate.

Chongruksut et al. [19], based on the evaluation of six randomized trials of routine shunting versus selective shunting, concluded that the available data were too limited to support the superiority of one method. None of the monitoring methods for selective shunting gave better results. It seems that such conclusions induce research on methods that will assess the efficiency of collateral circulation to the brain in the preoperative period.

There are publications describing serious complications after shunt application [20,21]. A limitation of the above methods is their application after qualification and the beginning of the CEA, which limits the possibility of using endovascular treatment where the fact of CC intolerance is not so important.

Determining the risk of CC intolerance before surgery seems to be of much more practical importance. In patients with a small benefit from surgery, it may affect the decision to continue the optimal medical treatment alone [1], while in patients who have favorable conditions for endovascular treatment, to qualify for CAS.

Most of the publications refer to prediction of CC intolerance based on the anatomy of the circle of Willis [22,23]. Correlation between intracranial anomaly abnormalities and CC intolerance is very high [24]. It grows significantly after specifying the number of abnormal arteries and the nature of the lesions. However, the evaluation of anatomical factors alone results in a relatively high number of false positive results. This results in a significantly higher percentage of shunts being used [25]. In centers using shunts selectively the usage frequency exceeds 20% [26].

Our own observations and the literature data show that in the subgroup of patients with high CC intolerance the percentage of neurological complications during CEA is unacceptable [3,27]. In our department we withdraw from surgery and qualify patients for carotid stenting in such cases.

## 5. Conclusions

Implementation of anatomical and functional brain examinations increases the chance of predicting cross-clamping intolerance during carotid endarterectomy. The highest value in predicting clamping intolerance is achieved through a combined analysis of PA-MR and MRA of the circle of Willis. The important variable significantly related to CC tolerance is hypertension.

## Figures and Tables

**Figure 1 jcm-09-03913-f001:**
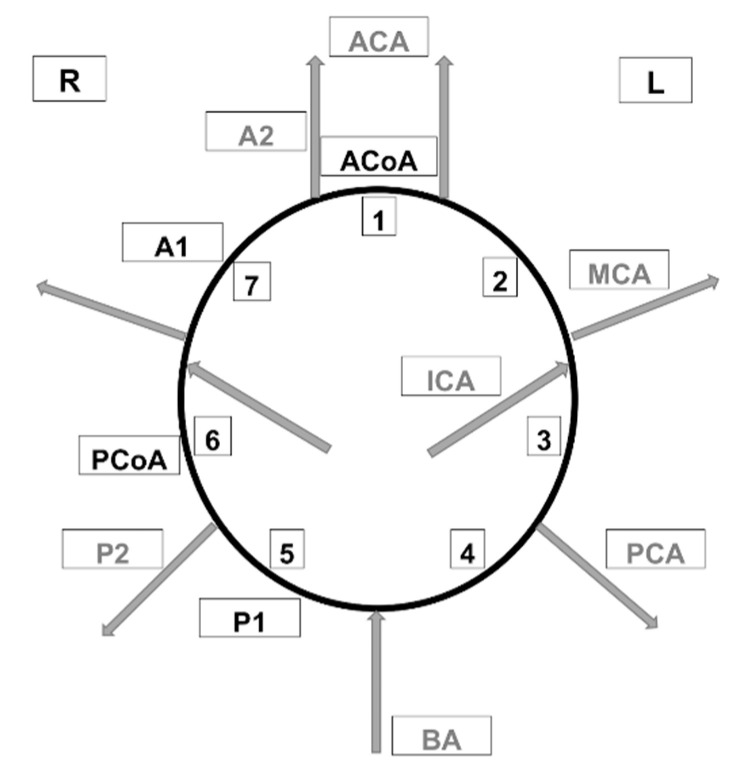
Arteries of the circle of Willis considered when calculating the W index. ACA—anterior cerebral artery, PCA—posterior cerebral artery, BA—basilar artery, AcoA—anterior communicating artery, PCoA—posterior communicating artery, MCA—middle cerebral artery, ICA—internal carotid artery, A1—1st segment of anterior cerebral artery, A2—2nd segment of anterior cerebral artery, P1—1st segment of posterior cerebral artery, P2—2nd segment of posterior cerebral artery.

**Figure 2 jcm-09-03913-f002:**
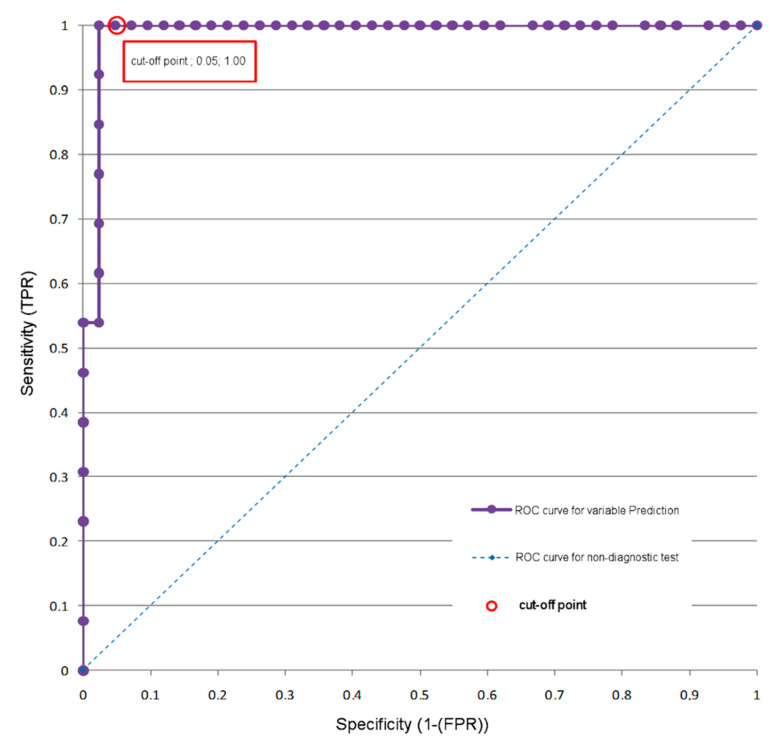
ROC analysis—a model using the magnetic resonance brain perfusion with acetazolamide and magnetic resonance angiography of the intracranial arteries. ROC curve for the prediction variable (by cross-clamping test results).

**Figure 3 jcm-09-03913-f003:**
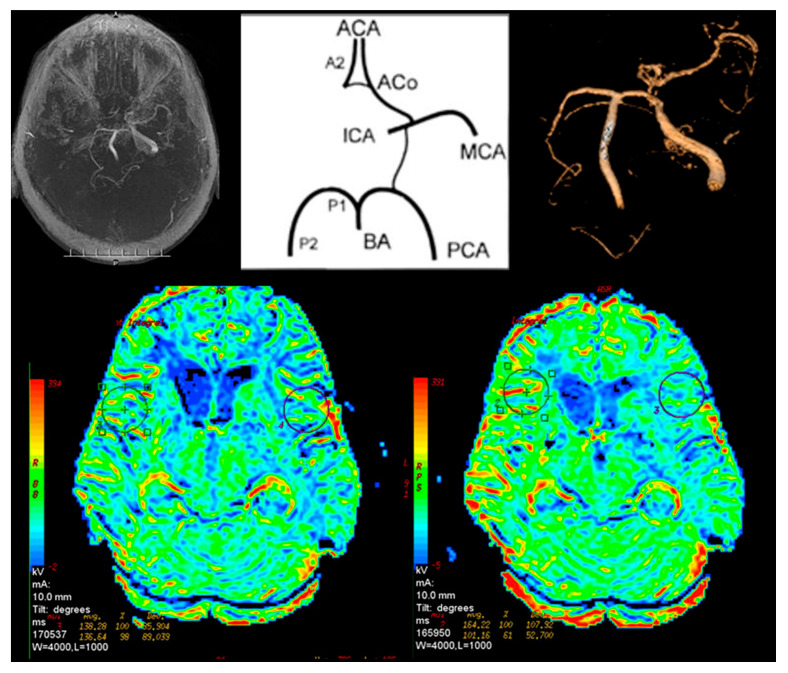
Magnetic resonance brain perfusion with Acetazolamide and magnetic resonance angiography of the intracranial arteries. Right internal carotid artery—100%, left internal carotid artery—70%. Differences after admission of Acetazolamide marked with circles (right side).

**Figure 4 jcm-09-03913-f004:**
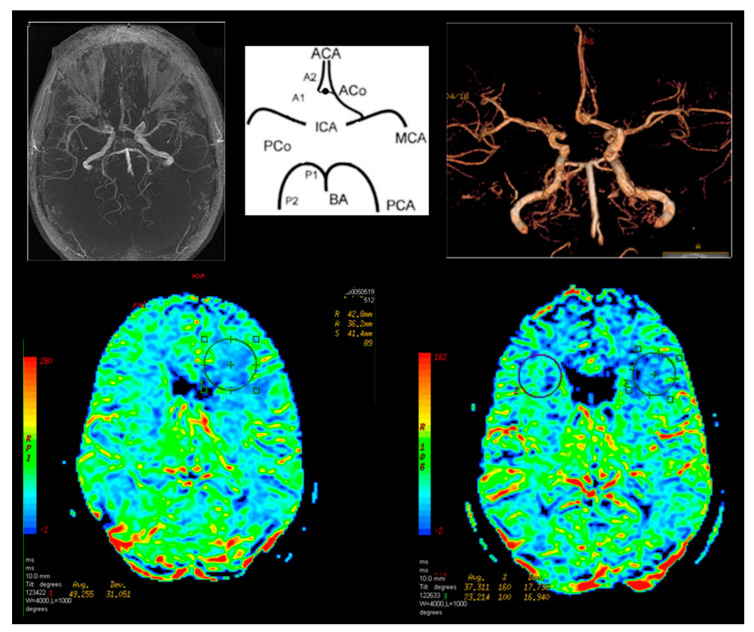
Magnetic resonance brain perfusion with Acetazolamide and magnetic resonance angiography of the intracranial arteries. Right internal carotid artery—70%, left internal carotid artery—90% (operated side). Differences after admission of Acetazolamide marked with circles (right side).

**Table 1 jcm-09-03913-t001:** Demographic and epidemiological data.

	Group I	Group II
Number of patients	40	15
Age (years)	51–84 (average 68.1)	66–83 (average 69.8)
Sex:	F-14 (35%) M-26 (65%)	F-5 (33.3%) M-10 (66.7%)
Comorbidities:		
Coronary disease	29 (72.5%)	10 (66.7%)
Myocardial infarction	7 (17.5%)	3 (20%)
Hypertension	38 (95%)	10 (66.7%)
Diabetes t.2	15 (37.5%)	5 (33.3%)
Obesitas	5 (12.5%)	2 (13.3%)
Stroke:		
Ipsilateral	4 (10%)	2 (13.3%)
Contralateral	5 (12.5%)	2 (13.3%)
TIA:		
Ipsilateral	15 (37.5%)	7 (46.7%)
Contralateral	5 (12.5%)	-
Asymptomatic	11 (27.5%)	4 (26.6%)
ICA stenosis-ipsilateral		
0–49%	0	0
50–69%	0	0
70–95%	40 (100%)	15 (100%)
95–99%	0	0
100%	0	0
ICA stenosis-contralateral		
0–49%	6 (15%)	2 (13%)
50–69%	22 (55%)	9 (60%)
70–95%	6 (15%)	2 (13%)
95–99%	0	0
100%	6 (15%)	2 (13%)

TIA—transient ischemic attack, ICA—internal carotid artery.

**Table 2 jcm-09-03913-t002:** Distribution of pathologies in the circle of Willis in Group I and Group II.

	A1	ACA	A2-A5	P1	PCoA	Carotisation	BA
**Group I**							
**Decrease of signal**	12	2	5	0	0	0	8
**Lack of signal**	21	12	10	1	29	5	1
**Group II**							
**Decrease of signal**	3	0	1	0	0	0	0
**Lack of signal**	0	0	1	0	8	0	0

A1—1st segment of anterior cerebral artery, ACA—anterior cerebral artery, A2–A5 2nd–5th segment of anterior cerebral artery, P1—1st segment of posterior cerebral artery, PCoA—posterior communicating artery, BA—basilar artery.

**Table 3 jcm-09-03913-t003:** CC intolerance and CC tolerance test in terms of variables being potential predictors of resistance to the CC. Comparison of Group I and Group II.

Variable	CC Intolerance		CC Tolerance		Significance of the Levene Test	Variance in Both Groups	T Statistics	df	Significance of the *t* Test (Two Sided)
	Average	Standard Deviation (SD)	Average	Standard Deviation (SD)					
**P-MR/nA**	10.5	5.2	6.3	3.2	0.078	Equal	2.77	53	**0.008**
**P-MR/A**	19.8	5.4	9.6	4	0.126	Equal	6.21	53	**0**
**W Index**	4.52	1.82	1.54	1.61	0.634	Equal	5.29	53	**0**
**Flow in A1**	1.33	0.79	0.15	0.38	0	Different	7.38	43.31	**0**
**Flow in ACoA**	0.62	0.91	0	0	0	Different	4.41	41	**0**
**Flow in BA**	0.238	0.484	0	0	0	Different	3.19	41	**0.003**
**Carotisation**	0.143	0.417	0	0	0.009	Different	2.22	41	**0.032**

A1—1st segment of anterior cerebral artery, ACoA—anterior communicating artery, BA—basilar artery, CC—cross-clamping, P-MR—magnetic resonance brain perfusion tests, P-MR/nA—difference in the P-MR in relation to the contralateral hemisphere without acetazolamide, P-MR/A—difference in the P-MR in relation to the contralateral hemisphere with acetazolamide.

**Table 4 jcm-09-03913-t004:** Sensitivity and specificity of CC calculated for a cut-off point of 0.322.

	CC Tolerance		CC Intolerance	Summary
**Tolerance (prognosis)**		**TP**	**FP**	
	N	13	2.1	15.1
	%	100%	5%	
**Intolerance (prognosis)**		**FN**	**TN**	
	N	0	39.9	39.9
	%	0%	95%	
**Summa** **ry**		13	42	55

TP—true positive (sensitivity), TN—true negative (specificity), CC—cross-clamping, FN—false negative, FP—false positive.

**Table 5 jcm-09-03913-t005:** Predictive power indicators depending on the analyzed range of data.

	MRA + P-MR	P-MR	MRA
**FN%**	0	15.4	45.4
**ACC%**	96.2	92.5	85.5
**R^2^%**	90.2	87.1	74.7

MRA—magnetic resonance angiography, P-MR magnetic resonance brain perfusion tests with Acetazolamide, FN—false negative, ACC—accuracy.

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
