# Peer review of "The Anatomy of the Circle of Willis Is Not a Strong Enough Predictive Factor for the Prognosis of Cross-Clamping Intolerance during Carotid Endarterectomy"

_jcm, 2020, doi:10.3390/jcm9123913_

Round 1
Reviewer 1 Report
This investigation is of relevance for carotid surgery, since it is still unknown how to predict the necessity of surgery with a temporary intralumal shunt. To underminde their thesis, a mathematical model was used (ROC curve) with definition of a cutoffpoint.
There are a few questions to be answered in the manuscript and minor changes recommended:
Questions:
Why was group II only so mall (15 pts) - the majority of patients has no cross-clamping intolerance, it would have been possible to match the patients.
On page 2 it is mentioned, that in some patients (number unclear) the operation was cancelled in the case of clamping ischemia. Did these patients receive a shunt ( takes about 2 minutes) before the operation was cancelled?
Was there a difference in the predictive results for symptomatic and asymptomatic patients? Was there a difference in the case of contralateral occlusion and/or hypertension? (Maybe a larger cohort would be necessary, this should be at least discussed).
Changes:
Tabel 1 should include the numbers for contralateral ICA occlusions, and degree of stenosis (only mentioned in the text).
Discussion: 2 papers should be added in the references list.
The last Cochrane Review 2014:CD000190 should be cited and discussed.
Aburahma AF (J Vasc Surg 2011; 54(5):1502-10 contains an important literature review and should be cited and discussed. There is only a little difference in outcome of routine shunting and selective shunting (1,4% vs. 1.6% stroke rate), which limits the clinical relevance a little bit.
Author Response
Response to Reviewer 1 Comments
Point 1. Why was group II only so mall (15 pts) - the majority of patients has no cross-clamping intolerance, it would have been possible to match the patients.
Response 1: I agree that Group II, which includes 15 patients, is small. This was due to the financial limitations of the study.
We do not perform perfusion examinations as standard in patients qualified for CEA. After evaluating the results in 15 patients with cross-clamping tolerance, we found that the percentage of deviation was small and allowed us to draw conclusions.
We wanted to be able to perform tests on as many patients with cross-clamping intolerance as possible. In this group, the differences in anatomical/perfusion results were significantly greater and required a larger group.
p 4: 182-185
Point 2. On page 2 it is mentioned, that in some patients (number unclear) the operation was cancelled in the case of clamping ischemia. Did these patients receive a shunt (takes about 2 minutes) before the operation was cancelled?
Response 2: In everyday practice, we obviously use shunts in patients with cross-clamping intolerance. In the case of using a shunt, we do not cancel the operation, even in the event of the appearance of symptoms of cerebral ischemia. However, in our centre, in this small subgroup of patients, the rate of neurological complications is higher than in the general post-CEA population.
Our observations show that in the case of extreme intolerance, when symptoms of cerebral ischemia occur after a few seconds, a much safer option than using a shunt is conversion to CAS (direct access), or delayed stenting.
All patients who did not continue surgery (clamping without arteriotomy) were included in the study. The final assessment was performed for those who had all the tests performed according to the protocol.
Point 3. Was there a difference in the predictive results for symptomatic and asymptomatic patients? Was there a difference in the case of contralateral occlusion and/or hypertension? (Maybe a larger cohort would be necessary, this should be at least discussed).
Response 3: The analysis of the results aimed at identifying the predictors of the occurrence of symptoms of intraoperative cerebral ischemia indicated by the 7 most important factors. They can be divided into 2 groups depending on the type of imaging tests:
1) variables related to magnetic resonance angiography
2) variables related to brain perfusion.
The first group includes the following important factors:
- flow rate in the A1 segment of the anterior cerebral artery
- flow rate in the ACoA
- flow rate in the basilar artery
- occurrence of a developmental variety in the form of carotization of the posterior cerebral arteries
- W index (determining the measure of intolerance to cross-clamping depending on the number and degree of varieties and pathologies in the arteries forming the Willis circle).
In the second group, the following factors turned out to be significant:
- difference in brain perfusion in the ispilateral/contralateral hemisphere in the study without Acetazolamide,
- difference in brain perfusion in the ispilateral/contralateral hemisphere in the study with Acetazolamide.
In addition to the factors mentioned above (classified into 2 groups), additional variables emerged (using a regression model for: perfusion, MR angiography and both tests together), additional variables that were predictors of resistance to cross-clamping: hypertension and type II diabetes.
The predictors mentioned above play an important role in the next area of ​​investigation aimed at determining the results of imaging and perfusion tests in helping to predict the resistance to cross-clamping. The question was asked about the optimal set of variables, and at the same time about the predictive power of each of the selected variables, which explains the resistance to cross-clamping accurately. Three separate predictive models were developed with the assumption that we could use the results of: 1) MR perfusion and angiography, 2) CNS perfusion only and 3) MR angiography only. Carrying out 3 separate analyses was guided by the fact that it isn't possible to perform all the examinations mentioned above in every centre (hospital), as well as an increase in costs incurred in multiplying research.
It turned out that when analyzing the results of perfusion tests and magnetic resonance angiography together, the most important predictor of resistance to cross-clamping is the result of CNS perfusion with Acetazolamide. Secondly, the results of MR angiography and the W index calculated on their basis should be taken into account. The last, third variable, significantly related to the resistance to the clamp test is the presence of arterial hypertension.
The analyses conducted did not show a significant effect on resistance to cross-clamping of factors such as cotralateral stenosis of ICA and the presence of neurological symptoms in the preoperative period. The study groups were small. We treat the results as a trend and understand that a large cohort would be necessary.
p 9: 285-288
Changes:
- Table 1 was supplemented with data on ipsilateral and contralateral ICA stenosis (p.5)- as suggested by the reviewer.
- I have added the proposed Aburahma article as [18] reference and a supplement to the discussion: p. 9: 224-232
[18] Aburahma AF, Mousa AY, Stone PA. Shunting during carotid endarterectomy.
J Vasc Surg. 2011;54(5):1502-10.
p.11-326
- I have added the proposed Chongruksut article as [19] reference and a supplement to the discussion: p. 9: 233-237
[19] Chongruksut W, Vaniyapong T, Rerkasem K. Routine or selective carotid artery shunting for carotid endarterectomy (and different methods of monitoring in selective shunting). Cochrane Database Syst Rev. 2014(6): Art. No.: CD000190. doi: 10.1002/14651858.CD000190.pub3.
p.11-328

Reviewer 2 Report
The topic has already been studied in the past, so no novelty is present herein.
There are two big flaws:
*Number of patients (40 vs. 15)
*How did they manage the choice of the control group (propensity matching? if this is the case, just 15 cases out of 883 CEAs?)
From my personal point of view, which is exclusively clinical, I didn't find easy-to-read either take-home messages for operating surgeon or radically new information regarding the relationship between new preop evaluation-CCC intolerance-ability to protect/reduce the risk of postoperative risk of stroke which is the real problem in such cohort of patients. The fact that these Authors did find hypertension to be a predictor for CCC intolerance confirms the fact that this event may not be further optimized by sophisticated though elegant preoperative evaluations, and owing to the fact that we do not have the entire cohort of CEA evaluated these findings indeed need to be re-evaluated in a prospective larger larger cohort.
Author Response
Response to Reviewer 2 Comments
Point 1. *Number of patients (40 vs. 15)
*How did they manage the choice of the control group (propensity matching? if this is the case, just 15 cases out of 883 CEAs?)
Response 1. Group II, which includes 15 patients, is small. This was due to the financial limitations of the study. Patients in the group were not selected randomly. It was 15 patients with cross-clamping tolerance operated on by a single surgeon (P.M.) who gave their informed consent to perform pre- and postoperative examinations.
p 4: 182-185
Point 2. From my personal point of view, which is exclusively clinical, I didn't find easy-to-read either take-home messages for operating surgeon or radically new information regarding the relationship between new preop evaluation-CCC intolerance-ability to protect/reduce the risk of postoperative risk of stroke which is the real problem in such cohort of patients. The fact that these Authors did find hypertension to be a predictor for CCC intolerance confirms the fact that this event may not be further optimized by sophisticated though elegant preoperative evaluations and owing to the fact that we do not have the entire cohort of CEA evaluated these findings indeed need to be re-evaluated in a prospective larger larger cohort.
Response 2. We do not perform perfusion examinations as standard in patients qualified for CEA. After evaluating the results in 15 patients with cross-clamping tolerance, we found that the percentage of deviation was small and this allowed us to draw preliminary conclusions.
We wanted to be able to perform tests on as many patients with cross-clamping intolerance as possible. In this group, the differences in anatomical/perfusion results were significantly greater and required a larger group.
When analyzing the results of perfusion tests and magnetic resonance angiography together, the most important predictor of resistance to cross-clamping is the result from using CNS perfusion with Acetazolamide. Secondly, the results of MR angiography and the W index calculated on their basis should be taken into account. The last, third variable, significantly related to the resistance to the cross-clamping is the presence of arterial hypertension (cannot be considered in isolation of course).

Reviewer 3 Report
Well designed manuscript with an interesting point. I am reluctant to what extent these two tools can be used for predicting CC tolerance in real-life clinical practice.
Author Response
Response to Reviewer 3 Comments
Point 1. Well designed manuscript with an interesting point. I am reluctant to what extent these two tools can be used for predicting CC tolerance in real-life clinical practice.
Response 1. The combination of anatomical and perfusion studies significantly increases the predictability of cross-clamping intolerance.
The use of such a testing scheme with each CEA is unlikely in daily practice due to cost and availability, especially in small centres.
However, there are groups of patients for which the practical application of research results brings measurable benefits, especially when the cross-clamping time is long and unpredictable. This applies to patients qualified for surgery due to carotid ICA aneurysms, neoplasms of the neck area, ICA stenosis in patients after radiotherapy with contraindications to CAS. It also applies to all those situations where an ICA ligation is being considered.
The number of such operations accounts for a small percentage of all carotid procedures and extended preoperative diagnostics in these cases is feasible in practice.

Round 2
Reviewer 2 Report
-